FOCUS: an alignment-free model to identify organisms in metagenomes using non-negative least squares

Silva Genivaldo Gueiros Z. 1
Cuevas Daniel A. 1
Dutilh Bas E. 4 5
Edwards Robert A. 1 2 3 5 6 redwards@mail.sdsu.edu
1 Computational Science Research Center, San Diego State University , San Diego, CA , USA
2 Department of Computer Science, San Diego State University , San Diego, CA , USA
3 Department of Biology, San Diego State University , San Diego, CA , USA
4 Centre for Molecular and Biomolecular Informatics, Radboud Institute for Molecular Life Sciences, Radboud University Medical Centre, GA , Nijmegen , The Netherlands
5 Department of Marine Biology, Institute of Biology, Federal University of Rio de Janeiro , Brazil
6 Division of Mathematics and Computer Science, Argonne National Laboratory , Argonne, IL , USA
Wang Yong
Electronic publication date: 2014 Jun 5
Publication date: 2014
Volume: 2
Electronic Location ID: e425
Received 2014 Mar 19; Accepted 2014 May 21
Copyright: © 2014 Silva et al.
Copyright year: 2014
Copyright holder: Silva et al.
License: This is an open access article distributed under the terms of the Creative Commons Attribution License, which permits unrestricted use, distribution, reproduction and adaptation in any medium and for any purpose provided that it is properly attributed. For attribution, the original author(s), title, publication source (PeerJ) and either DOI or URL of the article must be cited.
License URL: https://creativecommons.org/licenses/by/4.0/

Keywords: Metagenomes, Modeling, k-mer

Funding: NSF DEB-1046413 CNS-1305112 NWO Veni 016.111.075 CAPES/BRASIL Dutch Virgo Consortium GGZS and DAC were supported by NSF Grants (DEB-1046413 and CNS-1305112 to RAE). BED was supported by NWO Veni (016.111.075), CAPES/BRASIL and the Dutch Virgo Consortium. The funders had no role in study design, data collection and analysis, decision to publish, or preparation of the manuscript.

==============================
One of the major goals in metagenomics is to identify the organisms present in a microbial community from unannotated shotgun sequencing reads. Taxonomic profiling has valuable applications in biological and medical research, including disease diagnostics. Most currently available approaches do not scale well with increasing data volumes, which is important because both the number and lengths of the reads provided by sequencing platforms keep increasing. Here we introduce FOCUS, an agile composition based approach using non-negative least squares (NNLS) to report the organisms present in metagenomic samples and profile their abundances. FOCUS was tested with simulated and real metagenomes, and the results show that our approach accurately predicts the organisms present in microbial communities. FOCUS was implemented in Python. The source code and web-sever are freely available at http://edwards.sdsu.edu/FOCUS.

Introduction

Microbes are more abundant than any other cellular organism (Whitman, Coleman & Wiebe, 1998), and it is important to understand which organisms are present and what they are doing (Handelsman, 2004). In many environments a majority of the microbial community members cannot be cultured, and metagenomics is a powerful tool to directly probe uncultured genomes and understand the diversity of microbial communities by using only their DNA (Sharon & Banfield, 2013).

Understanding microbial communities is important in many areas of biology. For example, metagenomes can distinguish taxonomic and functional signatures of microbes associated with marine animals (Trindade-Silva et al., 2012) or disease states (Belda-Ferre et al., 2012). Large sequencing volumes, short read lengths, and sequencing errors make the task of identifying the diversity of organisms present in metagenomes challenging (Mande, Mohammed & Ghosh, 2012). Many programs exist for this and they are either homology- or composition-based.

Homology-based programs normally use the BLAST program (Altschul et al., 1997) to identify the best hit in a large database output. In MG-RAST (Meyer et al., 2008) sequences are aligned to a set of databases in order to classify the metagenomic sample. MetaPhlAn (Segata et al., 2012) and GenomePeek (K McNair, R Edwards, unpublished data) use a reduced database containing only marker genes, e.g., unique clades and housekeeping genes, allowing the BLAST search to be fast. PhymmBL (Brady & Salzberg, 2011) improves the BLAST results using interpolated Markov models. GASiC (Lindner & Renard, 2013) uses Bowtie (Langmead et al., 2009) and the reference genomes similarities to correct the observed abundance estimated. Parallel-Meta (Su, Xu & Ning, 2012) a fast program, which requires a GPU, uses megaBLAST (Zhang et al., 2000) and HMM (Hidden Markov Model) to improve the homology result. Most of these applications classify sequences individually, and generate a taxonomic profile by summing the bins.

In general, composition-based approaches use oligonucleotide (k-mer) frequencies. Taxy (Meinicke, Aßhauer & Lingner, 2011) uses oligonucleotide distributions in metagenomes and in reference genomes and uses mixture modeling to identify the organisms present in the metagenome, and RAIphy (Nalbantoglu et al., 2011) identifies organisms using oligonucleotides and relative abundance index.

We developed a new approach that reconstructs a taxonomic profile using an ensemble k-mer composition of the entire metagenome. We compute the optimal set of organism abundances using non-negative least squares (NNLS) to match the metagenome k-mer composition to organisms in a reference database. k-mers have previously been used to cluster unknown sequences (Teeling et al., 2004; McHardy et al., 2007) and NNLS has been used to identify the genera present in metagenomic samples based on variations in gene count (Carr, Shen-Orr & Borenstein, 2013). Here we combine these two approaches in FOCUS, an ultra fast, accurate, composition based approach to identify the taxa present in a metagenome. We compare the performance of FOCUS to GASiC, MetaPhlAn, RAIphy, PhymmBL, Taxy, and MG-RAST.

Methods

FOCUS workflow is described in Fig. 1. As in most composition-based approaches, a training set is pre-generated using the complete genomes information, and here the non-negative least squares (NNLS) is applied to compute the relative abundance of each organism in the database into the unknown data.

Figure 1 Workflow of the FOCUS program.

Reference dataset

FOCUS requires a group of reference genomes to model and identify the organisms present in a metagenome. 2,766 complete genomes were downloaded from the SEED servers (Aziz et al., 2012) on 20 December 2013 (see Table S1). k-mer frequencies (k = 6–8, default: k = 7) were calculated for both strands using Jellyfish 1.1.6 (Marçais & Kingsford, 2011), reducing the number of dimensions (Strous et al., 2012), and k-mer counts were normalized by the sum of frequencies. The user can also create their own training set, which is scalable to the quickly increasing number of available reference genomes because it also uses Jellyfish in the k-mer counting.

Simulated and real metagenomes

In order to evaluate FOCUS performance, a simulated dataset of short sequences (SimShort), containing 500,000 single 100 nt reads, was created using the supplied error model for Illumina GA IIx with TrueSeq SBS Kit v5-GA using GemSim (McElroy, Luciani & Thomas, 2012) (Table S2). The previous published high complexity simulated dataset (SimHC) from FAMeS (Mavromatis et al., 2007) was also used in the evaluation. Moreover, real metagenomic datasets were selected as test cases: one under healthy conditions, one under disease conditions (MG-RAST accession 4447943.3 and 4447192.3) (Belda-Ferre et al., 2012), one fecal sample from a healthy individual (MG-RAST accession 4440945.3) (Kurokawa et al., 2007), and three hundred datasets from the Human Micriobiome Project (HMP) (Consortium, 2012) (Table S3) were selected as a test case.

Non-negative least squares (NNLS)

The estimation of a parameterized model to understand some data is a fundamental problem in data modeling. Nevertheless, the estimation is not always easy, e.g., in problems like metagenome profiling that cannot have negative values for the fitted parameters. In such case, a solution can be estimated using NNLS, which is defined as:

Given a matrix A∈ℝmxn and a vector b∈ℝm, where m ≥ n, find a non-negative vector x∈ℝn to minimize the function (1). (1) fx=1/2∥Ax−b∥2,wherex≥0and∑i=inxi=1.

In FOCUS, the reference matrix A is composed of m k-mer frequencies from n genomes, while a vector describing the user’s metagenomic dataset is calculated from the k-mer frequencies of both strands from the dataset using Jellyfish. FOCUS uses non-negative least squares to compute the set of k-mer frequencies x that explains the optimal possible abundance of k-mers in the user’s metagenome by selecting the optimal number of frequencies from the matrix A. We minimize the sum of squared differences (1) using the open source Scipy library (Jones, Oliphant & Peterson, 2001) which has a module for the NNLS algorithm which solves the KKT (Karush–Kuhn–Tucker) conditions (Lawson & Hanson, 1974). We added Tikhonov regularization (Garda & Galias, 2012) to deal with genomes that have similar k-mer compositions.

Jackknife resampling of the data

We implemented a jackknife resampling strategy to assess the robustness of the results. 50% of the reads were randomly resampled 1000 ×, and the species frequencies recalculated. For each species, these 1000 frequencies were averaged and the standard deviation calculated to estimate the spread.

Web-based and graphical user interface version

As an alternative to the command line version of the program, we have created a user-friendly web version and a graphical user interface (GUI) for Microsoft Windows users. The web server and the GUI are available at http://edwards.sdsu.edu/FOCUS.

Results and Discussion

Evaluation and comparison with other tools

All tools were run using default parameters and their default reference database, either online (MG-RAST) or using one core on a server with 24 processors × 6 cores Intel(R) Xeon(R) CPU X5650 @ 2.67 GHz and 189 GB RAM. We only compared GASiC to the SimHC dataset which had the results previously published (Lindner & Renard, 2013). We tried to run the tool; however, it requires a large amount of storage during the process to save its output data.

For the real data, three hundred and three metagenomic datasets were selected. First, the metagenomic sample of the human oral cavity from diseased conditions was used. MetaPhlAn apparently over predicted the genera Veillonella due to the short genome, and Taxy did not predict Prevotella hits (see Fig. 2) as described in Belda-Ferre et al. (2012). FOCUS was able to profile the organisms in only 41 s. Taxy took about 45 s, MetaPhlAn took about 3 min, RAIphy took 52 min, MG-RAST took 3 days, and PhymmBL took 1 week and 6 days. Using random subsets for the oral metagenome, we tested the tools scalability and showed that FOCUS and Taxy profile metagenomes in constant time (see Fig. 3).

Figure 2 Genera-level taxonomy classification sorted by FOCUS prediction for the metagenome from a diseased human oral cavity using FOCUS, MetaPhlAn, MG-RAST, PhymnBL, RAIphy, Taxy, and FOCUS (mean).

Error bars represent the standard deviation uncertainty in tested metagenome.

Figure 3 Scalability test using different sub-sets of the human oral cavity under disease metagenome using FOCUS, MetaPhlAn, MG-RAST, PhymnBL, RAIphy, Taxy.

The oral metagenome from the healthy condition was used. MetaPhlAn possibly over predicted the genera Neisseria, and Taxy was not able to predict Rothia hits (see Fig. 4). FOCUS profiled the metagenome in only 35 s. Taxy took about 41 s, MetaPhlAn took about 2 min, RAIphy took 48 min, MG-RAST took 3 days, and PhymmBL took 9 days.

Figure 4 Genera-level taxonomy classification sorted by FOCUS prediction for the metagenome from a healthy human oral cavity using FOCUS, MetaPhlAn, MG-RAST, PhymnBL, RAIphy, Taxy, and FOCUS (mean).

Error bars show the standard deviation for the real metagenome.

A fecal metagenome from a healthy individual was used. All the tools predicted that Bifidobacterium and Enterococcus were the two most abundant genera in the sample. However, RAIphy apparently under predicted the genera Bifidobacterium (see Fig. 5). For this small dataset, FOCUS profiled the metagenome in 35 s. Taxy took about 40 s, MetaPhlAn took only 30 min, RAIphy took about 4 min, MG-RAST took 3 days, and PhymmBL took 2 days and 14 h.

Figure 5 Genera-level taxonomy classification sorted by FOCUS prediction for the metagenome from a fecal metagenomic sample of a healthy human using FOCUS, MetaPhlAn, MG-RAST, PhymnBL, RAIphy, Taxy, and FOCUS (mean).

Error bars show the standard deviation for the real metagenome.

Three hundred metagenomic samples (254 GB total) from HMP were analyzed at all the taxonomy levels using FOCUS (Table S4) in about 1 h and 20 min and compared with the published results from MetaPhlAn’s paper (Segata et al., 2012) by calculating the Euclidean distance between the results (see Fig. 6). For most of the samples, FOCUS and MetaPhlAn have similar predictions at the genera level but vary at the species level. However, for some samples in the posterior fornix and most of the samples from the anterior nares there were differences at all levels which may reflect the additional genome sequencing of isolates from those passages that has occurred since 2012. Other tools were not included in the analysis due to the CPU processing time.

Figure 6 Heat-map representing the distance between the FOCUS and MetaPhlAn results for 300 metagenomes from the Human Microbiome Project across 15 body sites.

The distance was computed using the Euclidean distance between the results of both tools.

For the simulated data, we removed species from the reference dataset that are present in this dataset and tried to predict the genera present in the SimShort dataset. A major limitation of many of the approaches discussed here is that the underlying databases cannot be changed. Only FOCUS, RAIphy, GASiC, and PhymmBL allow the end user to change their reference database. FOCUS and PhymmBL best predicted the correct genera while RAIphy could not correctly predict their abundance (Fig. 7). FOCUS had the fastest performance (45 s); RAIphy took about 2 h, while PhymmBL took approximately 5 days. Figs. S1–S5 show the same comparison for other taxonomy resolutions.

Figure 7 Genera-level taxonomy classification for the SimShort dataset using FOCUS, PhymnBL, RAIphy, and FOCUS (mean).

For the SimHC simulated metagenomes, the genera present in the dataset were deleted from the training dataset, and we evaluated the class-level prediction. The tested tools correctly predicted the classes, except that RAIphy over predicted the top two classes (see Fig. 8). Again, FOCUS was the fastest tool (30 s) in comparison to RAIphy, which took about 1 h and 50 min, and PhymmBL, which took about 4 days. See Figs. S6–S8 for other taxonomy levels.

Figure 8 Class-level taxonomy classification for the SimHC dataset using FOCUS, PhymnBL, RAIphy, and FOCUS (mean).

Furthermore, for the SimHC dataset, we ran all the previously used tools and the GASiC published results to evaluate the genera-level prediction. GASiC and PhymmBL had the best predictions, and FOCUS failed in the prediction of 4 minor genera probably because many organisms present in the SimHC dataset were not included in the FOCUS database (see Fig. 9). We did not compare the running time because we extracted the GASiC results from its paper; however, in the original paper it took 2 days and needed at least 500 GB of storage to analyze the SimHC simulated metagenome.

Figure 9 Genera-level taxonomy classification for the SimHC dataset using FOCUS, MetaPhlAn, MG-RAST, PhymnBL, RAIphy, Taxy, GASiC, and FOCUS (mean).

The very small standard deviations observed after jackknife re-sampling indicate the robustness of our results. Furthermore, in order to show a quantitative evaluation between the real and predicted abundance for the synthetic metagenomes, we computed the Euclidean distance between the real and predicted abundances for all the simulated data presented above (see Fig. 10). For some of the tools, only genus level predictions are available, but for RAIphy, PhymmBL, and FOCUS we included all taxonomic levels. The data demonstrate that FOCUS had the best prediction in more than half of test cases.

Figure 10 Numerical evaluation between the real and predicted abundance for the synthetic metagenomes computed by the Euclidean distance between the real and the predicted values.

These tests were performed on a server; however, FOCUS is also ultra fast on a simple computer. For example, we profiled the real dataset in 1 min and 45 s using an Intel(R) Core(TM) i3 @2.53 GHz and 1 GB RAM. In addition to the Web server, we have developed a stand-alone version that runs on the Windows® platform.

Limitations

As with other methods created to profile metagenome sequences, FOCUS depends on a curated database of microbial reference genomes in order to predict a specific genus. If a reference genome is absent, the tool will predict the closest reference available.

Conclusions

Here we present FOCUS, an agile solution to identify the organisms present in metagenomic samples that does not rely on mapping individual reads, but instead determines the taxonomic composition of the entire metagenome at once by using NNLS. This makes FOCUS an extremely fast and scalable solution to profile the focal taxa in a metagenome. FOCUS reports very similar species compositions as currently available, state of the art metagenome profiling tools.

Availability and requirements

Project name: FOCUS

Project and web server home page: http://edwards.sdsu.edu/FOCUS

Operating system: the program has a command line version that works on OS X and Unix, and a GUI for Microsoft Windows users.

Programming language: Python 2.7.

Other requirements: Numpy (http://www.numpy.org), Scipy (http://scipy.org), Jellyfish (http://www.cbcb.umd.edu/software/jellyfish), and Python programming language (http://www.python.org).

License: GNU GPL3.

Any restrictions to use by non-academics: no special restrictions.

Supplemental information

Table S1 Complete list of complete genomes present in the training set

Click here for additional data file.

Table S2 Complete list of organisms and abundances for the SimShort test set

Click here for additional data file.

Table S3 Complete list of three hundred metagenomes from the Human Micriobiome Project (HMP) used as test set

Click here for additional data file.

Table S4 FOCUS prediction for in all the levels for 300 metagenomes from the Human Microbiome Project

Click here for additional data file.

Figures S1--S8 Supplementary_Data with supplementary results

Click here for additional data file.

We thank Dr. Peter Blomgren for help with the Advanced Numerical Analysis, Raul Maia Falcao for working on an alternative version to count k-mers, and the reviewers for their useful comments.

Additional Information and Declarations

Competing Interests

Author Contributions

Data Deposition

The authors declare there are no competing interests.

Genivaldo Gueiros Z. Silva conceived and designed the experiments, performed the experiments, analyzed the data, contributed reagents/materials/analysis tools, wrote the paper, prepared figures and/or tables, reviewed drafts of the paper.

Daniel A. Cuevas contributed reagents/materials/analysis tools, wrote the paper, prepared figures and/or tables, reviewed drafts of the paper.

Bas E. Dutilh and Robert A. Edwards contributed reagents/materials/analysis tools, wrote the paper, reviewed drafts of the paper.

The following information was supplied regarding the deposition of related data:

https://sourceforge.net/projects/metagenomefocus/.

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
