# Peer review of "FOCUS: an alignment-free model to identify organisms in metagenomes using non-negative least squares"

_PeerJ, doi:10.7717/peerj.425_

## Round 0.1 · original submission · Major Revisions

The authors are suggested to test their new method in additional datasets. Also the methods and results part should be enhanced.

A point to point response should be provided along with the revision.

Reviewer 1 ·

Basic reporting

Firstly, the litreaure review is insufficient, as the 16S rRNA based organism detection methods are not even mentioned. The author should at least cite one of these 16S rRNA reference-based methods. Also, for efficiency analysis, the authors are better cite some efficient metagenotic analysis methods such as Parallel-Meta (Su, BMC Bioinformatics, 2012).

Secondly, the organization of the "Results" section is strange: seems that the authors are better discribe results on simulated data, and then on real data.

THirdly, the annotations (such as "R", and also "m") for formula 1 is not clera.

Finally, there are quite some wording problem for this draft, such as "what thoese organisms are doing and who they are" (logically not the right order). Another example: there is a missing "." at the end of sub-section "Jackknife resampling of the data".

Experimental design

The description of the simulate datasets is blur: how many organisms simulated, and what relative proportions?

Also, how "using dufferent subset" is implemented, and how to avoid bias?

These are compeltely unclear from the experiment design part of this draft.

Validity of the findings

The results part lack complete quantitative comparison of the accuracy of various methods, and thus making it difficult to judge the validity of the finding.

Reviewer 2 ·

Basic reporting

The methods section is very short and does not seem to cover sufficiently in depth all steps of the methodology. For example, I suspect that several parameters need to be tuned, but there is no mention of them.

Experimental design

"No Comments"

Validity of the findings

The presented approach to taxonomically profile metagenomes seems interesting and fast. However, it has been applied on very few metagenomes (2 synthetic and 1 real) and it thus fail to convince that FOCUS is a valid alternative with respect to existing approaches. Given that the tool is very fast to run, it should be easy for the authors to run it on several real and large metagenomes (for example they can use the HMP dataset for which the profiling with other methods - including 16S sequencing - are already publicly available).

In addition to a large set of real (and possibly synthetic) metagenomes there are several other points that should be addressed to really validate the method:
- for both synthetic metagenomes accuracies at all taxonomic levels should be presented
- a quantitative value summarizing the accuracy of the tested tools on the synthetic datasets should be given (squared error, correlation with the real values...)
- MetaPhlAn estimates the relative abundance of organisms, the other tools estimate the fraction of reads coming from each organism. They thus differ when the size of the genomes in the metagenome is not constant. Veillonella have very short genomes (~2MB) and thus will have higher genome relative abundance than reads relative abundance. This should be made clear when presenting and commenting Figure 1
- The authors trained the system on 2766 genomes. How fast is the training process? Is it scalable to the quickly increasing number of available reference genomes (not at least 10k)?

Reviewer 3 ·

Basic reporting

It is an important and valuable problem in metagenomics to identify the organisms present in a microbial community and estimate its abundance from unannotated sequencing reads. There have been a lot of methods on it. The authors argues that current methods do not scale well with increasing data volumes and they introduce a composition based approach using nonnegative least squares (NNLS) to estimate the focal organisms present in metagenomic samples and estimate their abundances. Generally, the idea is sound and brief tests have been done to demonstrate its effectiveness.

However, the paper is a bit too sketchy. Many details were not been covered by this manuscript. If possible, I would like to suggest the authors add more analysis and descriptions on their methods and results.

Moreover, another recent work has adopted a similar mathematical framework for the metagenomic abundance estimation. The authors should clarity the similarity and difference carefully.

Lindner MS, Renard BY. Metagenomic abundance estimation and diagnostic testing on species level. Nucleic Acids Res. 2013 Jan 7;41(1):e10. doi: 10.1093/nar/gks803.

Experimental design

The experimental design is reasonable, but the description is too sketchy.

Validity of the findings

The test is ok, but more analysis is needed.

---

## Round 0.2 · Major Revisions

Both reviewers think the current revision is not sufficient for publication. Some key points should be further addressed. In addition, the main contribution should be highlighted.

Reviewer 1 ·

Basic reporting

No Comments

Experimental design

The sample number in the experiment is insufficient to indicate the advantage of FOCUS: only 2 real and 2 simulated samples.

Validity of the findings

For the results of simulated data, the authors only illustrated the comparison among different tools in bar charts, without precise quantitative evaluation, such as Euclidean distance between the analysis results and the grand truth.

In the running time comparison, some software is web-based such as MG-RAST, which made the comparison to be unfair.

Additional comments

In this work the authors proposed an alignment free model named FOCUS for identifying organisms in metagenomes. However, I think this work is inadequate for publish in both software development and experiment design.
Major comments:
1. The core algorithm of FOCUS is implemented by others work such as Scipy library and Jellyfish.
2. The sample number in the experiment is insufficient to indicate the advantage of FOCUS: only 2 real and 2 simulated samples.
3. For the results of simulated data, the authors only illustrated the comparison among different tools in bar charts, without precise quantitative evaluation, such as Euclidean distance between the analysis results and the grand truth.
4. In the running time comparison, some software is web-based such as MG-RAST, which made the comparison to be unfair.

Minor comments;
There are some problems in the typesetting such as function (1) (line 102-103).

Reviewer 2 ·

Basic reporting

No Comments

Experimental design

No Comments

Validity of the findings

The authors revised the manuscript addressing some of my criticisms but several issues have not been solved.

The three outstanding main issues are reported below.

1. I remain convinced that applying a new approach on 2 synthetic metagenomes and 1 real one is not sufficient to provide evidence of its usefulness. Synthetic metagenomes are by definition artificial, so more real datasets should be analyzed. I partially agree with the fact that 16S sequencing datasets are not easily comparable with shotgun sequencing. However, I also mentioned that the application of other computational tools on shotgun metagenomic datasets are available (http://www.hmpdacc.org/HMSCP/ and http://www.hmpdacc.org/HMSMCP/) which gives the possibility of comparing FOCUS with respect to other validated tools on many real metagenomes without wasting CPU hours on running other tools.

2. All recent metagenomic profiling tools provide taxonomic resolution at the species level. I appreciate that the authors added some taxonomic levels, but species level results are the most important and biologically relevant ones and are still missing.

3. The authors might have misinterpreted my comment (and the same comment from Reviewer 1) about quantitative measure. I appreciate the evaluation strategy done on synthetic metagenomes; however, the results are shown only as barplots without providing numeric values for the comparison (and also leaving many clades in the "other" bin). For each specific clade (e.g. Streptococcus in Figure 5) the difference between the real and the estimated abundance for each tested method should be computed. Statistics about those differences (i.e. error) should be provided.

---

## Round 0.3 · accepted · Accept

The authors provided sufficient new data to further demonstrate the utility of FOCUS. In addition, extensive comparisons with existing methods are convincing. The manuscript has been significantly improved after this revision.